# Bioinformatics pipeline for the systematic mining genomic and proteomic variation linked to rare diseases: The example of monogenic diabetes

**Ksenia G. Kuznetsova**[1,2]*, **Jakub Vašíček**[1,2], **Dafni Skiadopoulou**[1,2], **Janne Molnes**[1,3], **Miriam Udler**[4,5,6], **Stefan Johansson**[1,3], **Pål Rasmus Njølstad**[1,7], **Alisa Manning**[4,5,6], **Marc Vaudel**[1,2,8]*

**1** Mohn Center for Diabetes Precision Medicine, Department of Clinical Science, University of Bergen, Bergen, Norway, **2** Computational Biology Unit, Department of Informatics, University of Bergen, Bergen, Norway, **3** Department of Medical Genetics, Haukeland University Hospital, Bergen, Norway, **4** Department of Medicine, Massachusetts General Hospital, Boston, MA, United States of America, **5** Metabolism Program, Broad Institute of MIT and Harvard, Cambridge, MA, United States of America, **6** Department of Medicine, Harvard Medical School, Boston, MA, United States of America, **7** Children and Youth Clinic, Haukeland University Hospital, Bergen, Norway, **8** Department of Genetics and Bioinformatics, Norwegian Institute of Public Health, Oslo, Norway

* ksenia.kuznetsova@uib.no (KGK); marc.vaudel@uib.no (MV)

**Data Availability Statement:** The data underlying the results presented in the study are available

## Abstract

Monogenic diabetes is characterized as a group of diseases caused by rare variants in single genes. Like for other rare diseases, multiple genes have been linked to monogenic diabetes with different measures of pathogenicity, but the information on the genes and variants is not unified among different resources, making it challenging to process them informatically. We have developed an automated pipeline for collecting and harmonizing data on genetic variants linked to monogenic diabetes. Furthermore, we have translated variant genetic sequences into protein sequences accounting for all protein isoforms and their variants. This allows researchers to consolidate information on variant genes and proteins linked to monogenic diabetes and facilitates their study using proteomics or structural biology. Our open and flexible implementation using Jupyter notebooks enables tailoring and modifying the pipeline and its application to other rare diseases.

## Introduction

The most common forms of monogenic diabetes are maturity-onset diabetes of the young (MODY), neonatal diabetes [1], inherited lipodystrophies, mitochondrial diabetes, among others [2]. Today, international guidelines are available for the diagnostic and follow-up of patients with suspected MODY [3]. These patients may now receive a molecular genetic diagnosis using diagnostic gene sequencing panels (https://panelapp.genomicsengland.co.uk/panels/472). This allows precise MODY subtyping and, depending on the diagnosis, the opportunity to avoid lifelong insulin medication and complications through lifestyle

from https://github.com/kuznetsovaks/MD_variants.

**Funding:** This research was funded, in whole or in part, by the Research Council of Norway (project #301178 to MV), the University of Bergen, and the Novo Nordisk Foundation (project NNF20OC0063872 to SJ). The publication of the manuscript was funded by Bergen Universitetsfond (2023/04/FOL to K.K.) A CC BY or equivalent license is applied to any Author Accepted Manuscript (AAM) version arising from this submission, in accordance with the grant's open access conditions. The funders had no role in study design, data collection and analysis, decision to publish, or preparation of the manuscript.

**Competing interests:** The authors have declared that no competing interests exist.

management or alternative treatment using oral antidiabetic drugs [4]. Furthermore, because the early correct diagnosis may implement successful treatment with low doses of sulfonylurea or diet alone and postpone complications, the timeline of diagnosis and care is thus crucial [4]. It is estimated that around 80% of monogenic diabetes cases remain undiagnosed due to symptomatic similarity to other types of diabetes [5]. These patients and their relatives remain unaware of their familial condition and do not benefit from adapted care.

The first challenge in establishing a firm diagnosis in all cases of MODY is the mapping of all genes that can cause monogenic diabetes. To date, multiple genes have been discovered to have associations with familial forms of diabetes [2]. Fourteen of them are notably often referred to as "the MODY genes" [3], although this list is subject to debate in the literature [6] and is being systematically assessed by international experts using established guidelines to determine gene-disease relationships (https://clinicalgenome.org/affiliation/40016). A second challenge is the difficulty in evaluating the pathogenicity of genetic variants [7], for which there is also an ongoing international effort to establish guidelines and provide expert variant curations in the ClinVar database (https://clinicalgenome.org/affiliation/50016). [8]. Furthermore, the response to alternative treatment might differ between populations [9]. One of the ways to address the challenge of precise diagnostics is to complement genetic screening with additional data, combining both molecular and clinical dimensions [10].

The recent advent of high-performance computational models for protein structures notably holds the promise to increase the throughput of the structural consequences of genetic variation [11]. Studying the protein sequences encoded by specific alleles can, for example, help understand whether their structure and properties are affected, hence shedding light on the pathogenicity of variants found during genetic diagnosis [12]. The adoption of these approaches is however impaired by the difficulty of mapping variants linked to rare diseases to the different forms of proteins that they encode. First, given that these variants are rare, the coverage by genomic databases is low. Maintaining an updated list of variants requires monitoring and mining of the literature by experts. Second, variants reported in the literature often lack standardization in their identifiers and coordinates, making it challenging to map them to a given genome build and requiring manual variant mapping. Third, inferring the consequences on protein products is still a daunting task for some variants (those alleles affecting splice sites and untranslated regions [UTRs], for example). Fourth, a given protein-coding sequence might encode different protein isoforms, which will produce different forms of proteins upon folding and post-translational modification [13], hence for a given variant multiple protein sequences need to be investigated. Mapping genetic variants linked to monogenic diabetes from genes to proteins is therefore not tractable and sustainable without automation using dedicated bioinformatic tools.

Here, we describe a new open-source modular pipeline based on Jupyter notebooks (https://eprints.soton.ac.uk/403913). that allows for the systematic collection of variants linked to monogenic diabetes and their mapping to Ensembl [14] and ClinVar [8]. We demonstrate how the different genes linked to monogenic diabetes harbor variants of different clinical significance. Finally, we translate the variant sequences to the protein level and provide the resulting sequences in a standard format that can readily be used for proteomic search and structural proteomic analyses using mass spectrometry or protein structure modeling.

## Methods

### General architecture

The pipeline consists of seven independent modules written in Python using Jupyter notebooks Fig 1. The notebooks are chained together as a pipeline, but they can also be used as

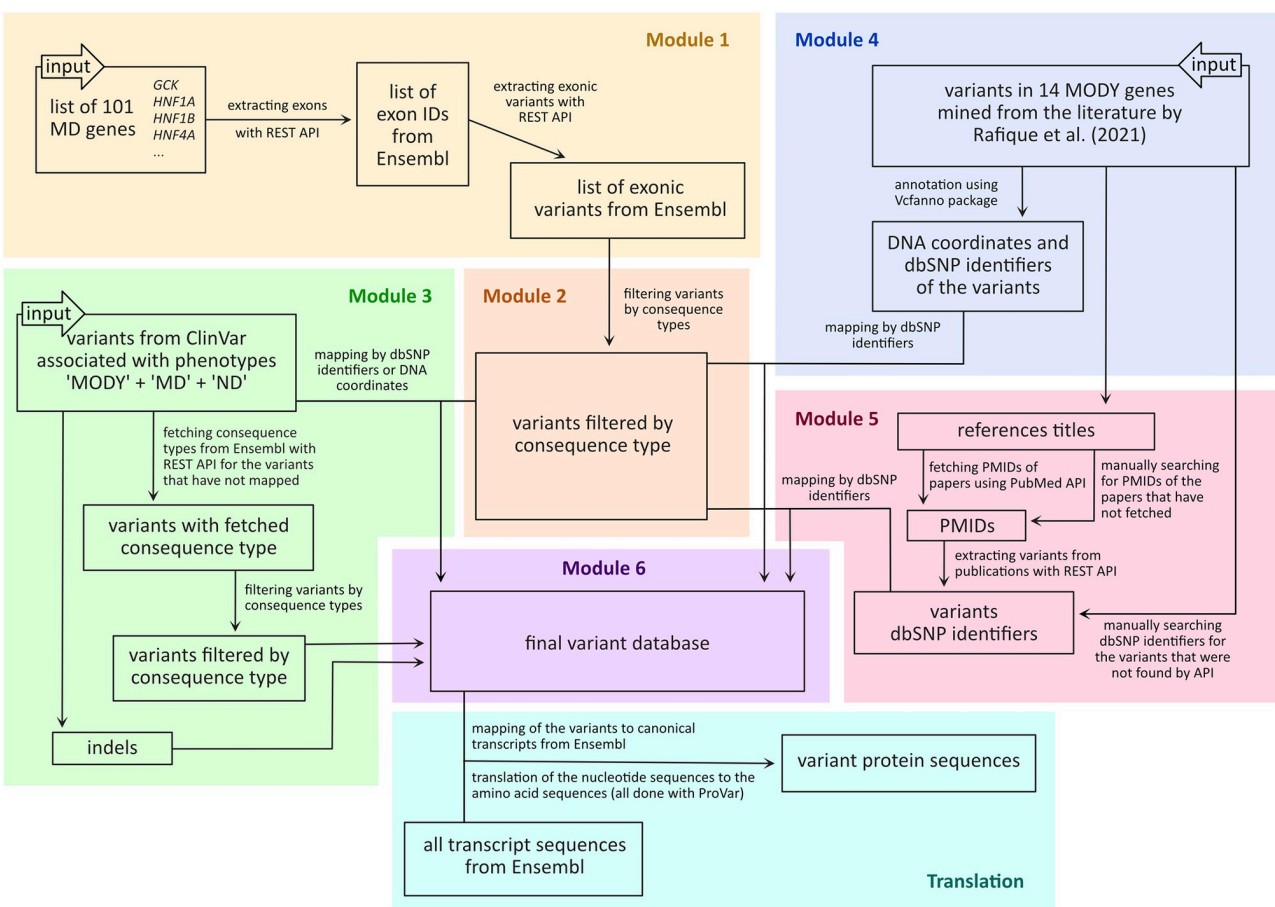

**Fig 1. General architecture of the pipeline.** Module 1: extract variants from Ensembl affecting genes linked to monogenic diabetes. Module 2: filter variant by consequence on the protein. Module 3: extract variants from ClinVar affecting genes linked to monogenic diabetes. Modules 4 and Module 5: extract the variants from the literature using the mining by Rafique et al. Module 6: consolidate the variants in a single table. Translation: produce the possible variant protein sequences. MODY—maturity-onset diabetes of the young, MD—monogenic diabetes, ND—neonatal diabetes, API—application programming interface, PMID—identifiers of scientific publications from the PubMed database, dbSNP identifiers—identifiers of the genomic variants from the dbSNP database, that start from "rs".

standalone applications, or integrated into other pipelines. First, the pipeline takes a list of genes and extracts exonic variants from Ensembl retaining only those variants that are predicted to affect protein sequences. Next, the program integrates the variants from ClinVar and maps them to Ensembl. Similarly, variants are extracted from the literature, here using the literature mining by Rafique et al. [15], and mapped to Ensembl. Subsequently, the harmonized collection of variants is consolidated in a database stored in the form of a text file that can easily be parsed and reused. Finally, the table of variants is mapped to all the transcripts linked by Ensembl to the genes of interest to obtain protein sequences of all the possible isoforms encoded by these genes. In this last step, the DNA sequences are translated to amino acid sequences and stored as protein FASTA files and here the user can also filter them to the isoforms of interest. To visually inspect the results, a separate module allows overlaying all the variants that can possibly affect a given gene onto the corresponding amino acid sequences.

## Module 1—Mining variants in Ensembl

In Module 1, a list of genes is mapped to Ensembl (here and further version 108) genes, transcripts, and exon identifiers using the Ensembl REST API [16]. Subsequently, the Ensembl REST API is queried using the exon identifiers to return all variants in Ensembl overlapping with the corresponding regions along with their annotation (identifiers, coordinates, consequences, etc.) For multi-allelic variants, we treat every alternative allele as a different entry, and add it as a new line to the table produced. The reference table is given in the Supplementary materials, S1 Table.

## Module 2—Categorizing by consequence and pathogenicity

Module 2 takes as an input the table of exonic variants from Ensembl generated in Module 1. For each consequence type, the prevalence of variants with different levels of pathogenicity is computed and visualized as heat maps Fig 2A. The same is done for the data from ClinVar (version October 2022) with all the variants regardless of the associated phenotype Fig 2B.

By default, only variants whose pathogenicity is classified mostly as likely pathogenic or pathogenic are retained for further analyses: "missense_variant", "protein_altering_variant", "coding_sequence_variant", "frameshift_variant", "splice_donor_variant", "splice_acceptor_variant", "splice_donor_5th_base_variant", "start_lost", "stop_gained", "stop_lost", "inframe_deletion", "inframe_insertion".

## Module 3. Mapping of the variants from ClinVar

Module 3 takes as an input three tables exported from ClinVar containing all the variants returned after querying three phenotypes: "MODY", "monogenic diabetes", and "neonatal diabetes". ClinVar annotates variants with different levels of pathogenicity: "pathogenic", "likely pathogenic", "uncertain significance", "likely benign" or "benign". The variants presenting dbSNP (release 154) identifiers are passed to Module 2. The others are mapped to Ensembl using genomic coordinates and alleles and predicted consequences are obtained with the Ensembl Variant Effect Predictor (VEP) [17] called on all variants using the Ensembl REST API. The variants with predicted consequences according to Ensembl are filtered in the same way as in Module 2 and listed in S2 and S3 Tables in the Supplementary materials. The variants where no consequence type was returned were all either short deletions, short insertions, or other short fragment replacements. These variants are listed in S4 and S5 Tables in the Supplementary materials. This subset is directly passed to Module 6. Note that the full set of tables is stored on GitHub.

## Module 4. Mapping variants from the literature (step 1 of 2)

Module 4 takes variants linked to monogenic diabetes according to the literature to cover the variants where the association with monogenic diabetes is not yet consolidated in ClinVar or Ensembl. As input we used the variants mined in the review by Rafique et al. [15] and provided as a supplementary table in their publication. The module uses the Vcfanno [18] library to annotate the variants with DNA coordinates and the dbSNP identifiers where possible. The Ensembl REST API is queried as in Module 1, and the results are passed to Module 6. The resulting variants in this step are given in Supplementary S6 Table.

## Module 5. Mapping variants from the literature (step 2 of 2)

For the variants that could not be mapped automatically in Module 4, the title of the publication as obtained from Module 4 is queried against the Entrez Programming Utilities API

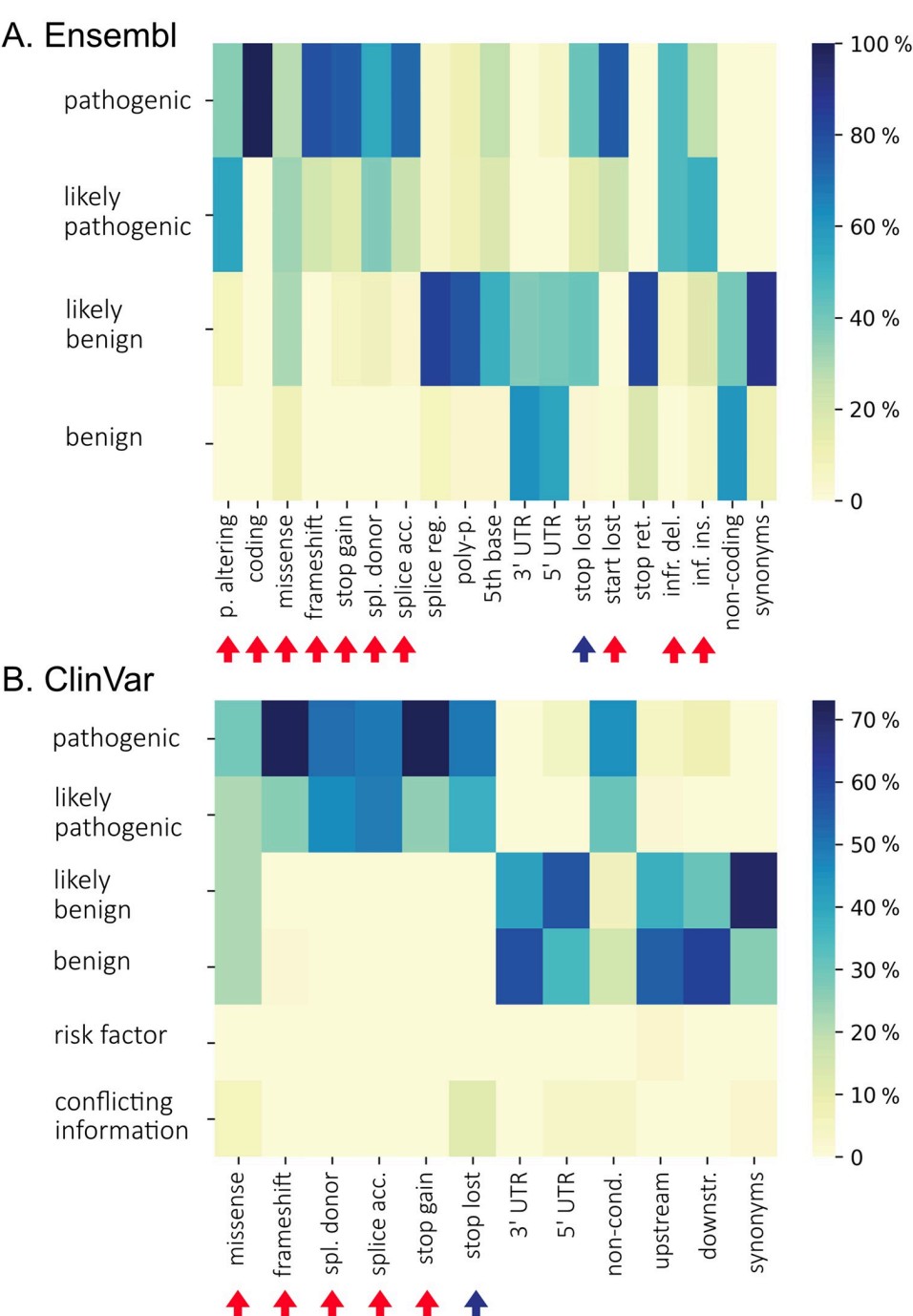

**Fig 2. Distribution of the consequence types within pathogenicity categories of the variants.** A: Variants from Ensembl, B: Variants from ClinVar. The heatmap shows the percentage of variants within pathogenicity categories for each consequence type. Arrows indicate the consequence types that were left in the Ensembl reference table created in Module 2 after filtering. "Stop lost" is marked with a blue arrow as it has conflicting evidence in two sources (see detailed in the text).

(https://www.ncbi.nlm.nih.gov/books/NBK25500). to return the PubMed (https://pubmed.ncbi.nlm.nih.gov). identifiers (PMIDs) of these articles. Note that some of the PMIDs were mapped and had to be added manually. Next, these PMIDs are used to query the same API and return the "rs" identifiers of the variants mentioned in these publications. Finally, the variants are mapped to Ensembl using their identifiers and passed to Module 6. Note that not all these variants could be mapped automatically and those that did not map to Ensembl were formatted manually for input to Module 6. Particularly, the variants from ClinVar that did not map and as a result do not have the "rs" identifiers were added to the table with the identifiers from ClinVar. Besides the identifiers, the table contains the chromosome number, the position of the variant, and the reference and alternative alleles. In total, 8 variants were added to VCF manually. The resulting variants in this step are given in Supplementary S7 Table.

## Module 6—Consolidation as table and VCF file

Module 6 combines all the tables produced by the previous modules and creates Venn diagrams showing the number of variants obtained from the different sources and their overlap. This table is then used to create a Variant Calling Format (VCF) file listing the site of all the variants.

We further categorized the variants into two levels of pathogenicity evidence. Level 1 variants include the variants from ClinVar obtained in Module 3 after querying three phenotypes: "MODY", "monogenic diabetes", or "neonatal diabetes" regardless of their reported clinical significance, complemented with the variants extracted from the literature in Module 4 and Module 5. Level 2 variants undergo stricter filtration criteria. For the variants obtained from ClinVar in Module 3, all variants labeled as benign, likely benign, or of uncertain significance were filtered out. For the variants obtained from the literature in Module 4 and Module 5, variants in *BLK*, *KLF11*, and *PAX4* were removed as these genes were reported to lack pathogenicity in MODY in more recent literature [6]. The output tables of this module are given as Supplementary S8 and S9 Tables for the 1st and the 2nd level variants correspondingly.

## Translation of the variant sequences into protein sequences

The translation step was performed using the ProVar tool (https://github.com/ProGenNo/ProHap). Shortly, the variants from the table obtained in Module 6 were mapped to canonical cDNA sequences from Ensembl to retrieve all the transcripts of the same gene with the annotation of start and stop codons. After mapping, all the sequences were translated to their amino acid sequences and written into a protein FASTA file. Both the table output and FASTA examples are given in Supplementary materials deposited on (https://doi.org/10.6084/m9.figshare.21444963.v2).

## Sequence overlay

All the variants from the resulting database were overlaid with the reference protein sequences obtained from Ensembl for all the transcripts of all genes. The variants are represented using two separate rows corresponding to the two levels of pathogenicity confidence (see example in Fig 3). FASTA files are parsed using the Pyteomics library [19]. The protein sequences and variants are plotted using the Matplotlib library [20].

## Results

Unlike more common forms of diabetes like type 2 diabetes (T2D), where large numbers of samples are available and federated initiatives consolidate information on genetic variants and their consequences in aggregated and harmonized forms (e.g. (https://t2d.hugeamp.org),

A. HNF1A protein sequence

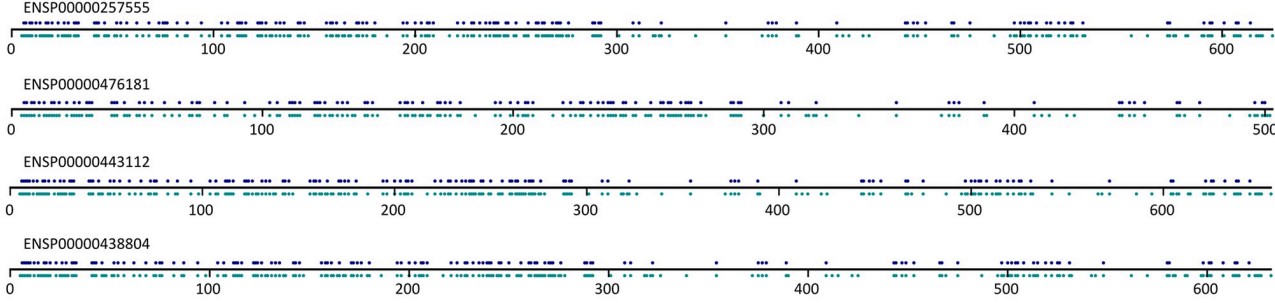

B. Zoomed in fragments of protein sequences of HNF1A and ABCC8

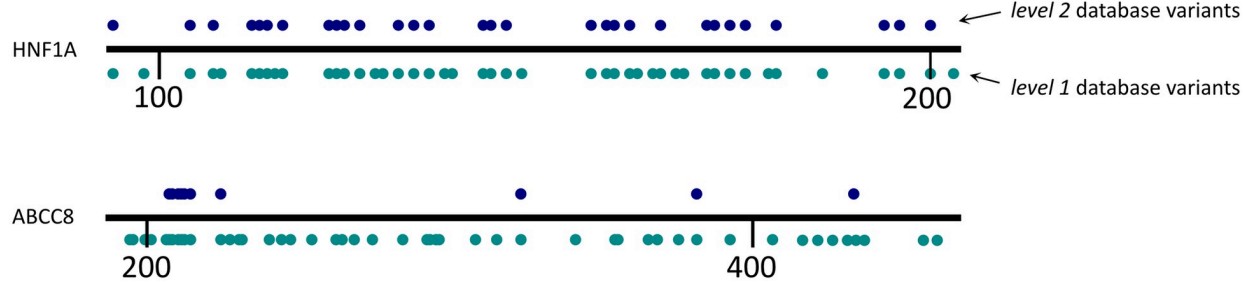

**Fig 3.** A: All isoforms of the protein product of *HNF1A* with the amino acid variants from the database mapped on the sequence. **B: Examples of the random fragments of the products of the canonical isoforms of *HNF1A* and *ABCC8* showing the difference in density of variants from level 1 and level 2 databases.** The top row of blue dots represents the positions of the variants from the level 2 database, whilst the bottom green row represents the positions of the variants from the level 1 database. Being the most well-studied gene, *HNF1A* has an almost equal number of dots in both rows as most of the reported variants are revised and confirmed as pathogenic. *ABCC8* has a lot of variants in the level 1 database that are not confirmed as pathogenic and, thus, are not in the level 2 database.

monogenic diabetes, as a rarer disease, relies on small cohorts and information on genetic variants is scattered in the literature and online databases. The aggregation and comparison of variants possibly linked with monogenic diabetes therefore currently relies on expert manual curation and annotation.

## Mining variants in genes linked to monogenic diabetes

We mined variants based on a list of 109 genes linked to monogenic diabetes, aiming at being as comprehensive as possible. These 109 genes consist of: i. 14 MODY genes taken from OMIM [21] or other reviews on MODY such as [3]; ii. 10 genes linked to neonatal diabetes, lipodystrophy, and insulin signaling taken from [2]; iii. 77 genes having variants with any evidence of association with either MODY, neonatal diabetes, or just the condition referred to as "monogenic diabetes" by ClinVar.

## Categorizing by consequence and pathogenicity

Since monogenic diabetes, being a Mendelian disease, is determined mostly by rare, highly penetrant coding variants [22], we focused on exonic variants when mining Ensembl.

Nevertheless, some variants reported in ClinVar and the literature mapped to untranslated regions (UTR) and splice regions. In these cases we decided whether to keep these variants or not based on the pathogenicity of variants in each consequence type category. To produce protein sequences, we focused on predicted consequences: "missense variant", "protein altering variant", or "coding sequence variant", and ignored variants on a transcript that were "noncoding transcript variant" or "synonymous variant". For other types of consequences, we focused on those enriched with pathogenic and likely pathogenic variants: "missense variant", "protein altering variant", "coding sequence variant", "frameshift variant", "splice donor variant", "splice acceptor variant", "stop retained", "stop gained", "inframe insertion", and "inframe deletion" (Fig 2). The "stop lost" consequences yielded different prevalence of pathogenic or likely pathogenic variants when considering the consequences reported by Ensembl vs. ClinVar (Fig 2), which can be explained by the differences between Ensembl and ClinVar. Here, the Ensembl dataset consists of the exonic variants, but is not linked to any pathological phenotype. The ClinVar dataset, though, was not narrowed down to any particular regions in the genes of interest, but all the variants in it are linked to clinical conditions. Therefore, the Ensembl dataset is enriched with protein coding variants, whereas the ClinVar dataset is enriched with pathogenic variants. The "stop lost" consequences were included in our analysis. Altogether, the resulting table variants contained 69,256 unique variants located in 109 genes.

## Categorizing by consequence and pathogenicity

We mapped 2,701 of these variants to variants linked to monogenic diabetes according to ClinVar or the literature, termed level 1 variants thereafter: 2,220 (82%) mapped uniquely to ClinVar, 136 (5%) to publications reviewed by Rafique et al., and 345 (13%) to both (Fig 4). An effect on protein sequences was predicted for 2,624 (97%) of them, resulting in 12,643 different protein sequences when accounting for all isoforms. The reason for some variants not reaching the final translated sequences is that some transcripts in Ensembl do not have an associated canonical protein product. These are included when selecting exons in Module 1 but do not yield protein sequences. On the other hand, genetic variation can cause translation of UTRs and other normally untranslated regions [23], and genetic variation in the UTRs and splice regions might affect the translation of the proteins in an indirect way [24, 25], but the effects of these variants on amino acid sequences remain challenging to predict. We further filtered the variants to retain only the pathogenic and likely pathogenic variants, termed level 2 variants thereafter, yielding 876 variants, of which 641 (73%) mapped uniquely to ClinVar, 160 (18%) to publications reviewed by Rafique et al., and 75 (9%) to both (Fig 4). An effect on protein sequences was predicted for 714 (82%) of them, resulting in 3,776 distinct protein sequences.

In both cases, the overlap between variants from ClinVar and the literature is limited. This can be explained by the fact that the ClinVar dataset consisted of the variants linked to all kinds of monogenic diabetes (109 genes) and the Rafique et al. dataset consisted only of the MODY variants (14 genes). Thus, we re-ran the pipeline considering only MODY variants from ClinVar. The overlap of the ClinVar MODY dataset was then 18% and 10% for the level 1 and 2 variants respectively. While the overlap was improved, it is still limited. This illustrates the importance to consider different sources of information when studying monogenic diabetes, and rare diseases in general.

It should be noted here that, in their work, Rafique et al. also used ClinVar as a source of variants. The variants from ClinVar that were not found by the literature text mining algorithm are listed in a separate supplementary file in their publication. We decided not to include this list in our work as we have included all the monogenic diabetes-associated variants from

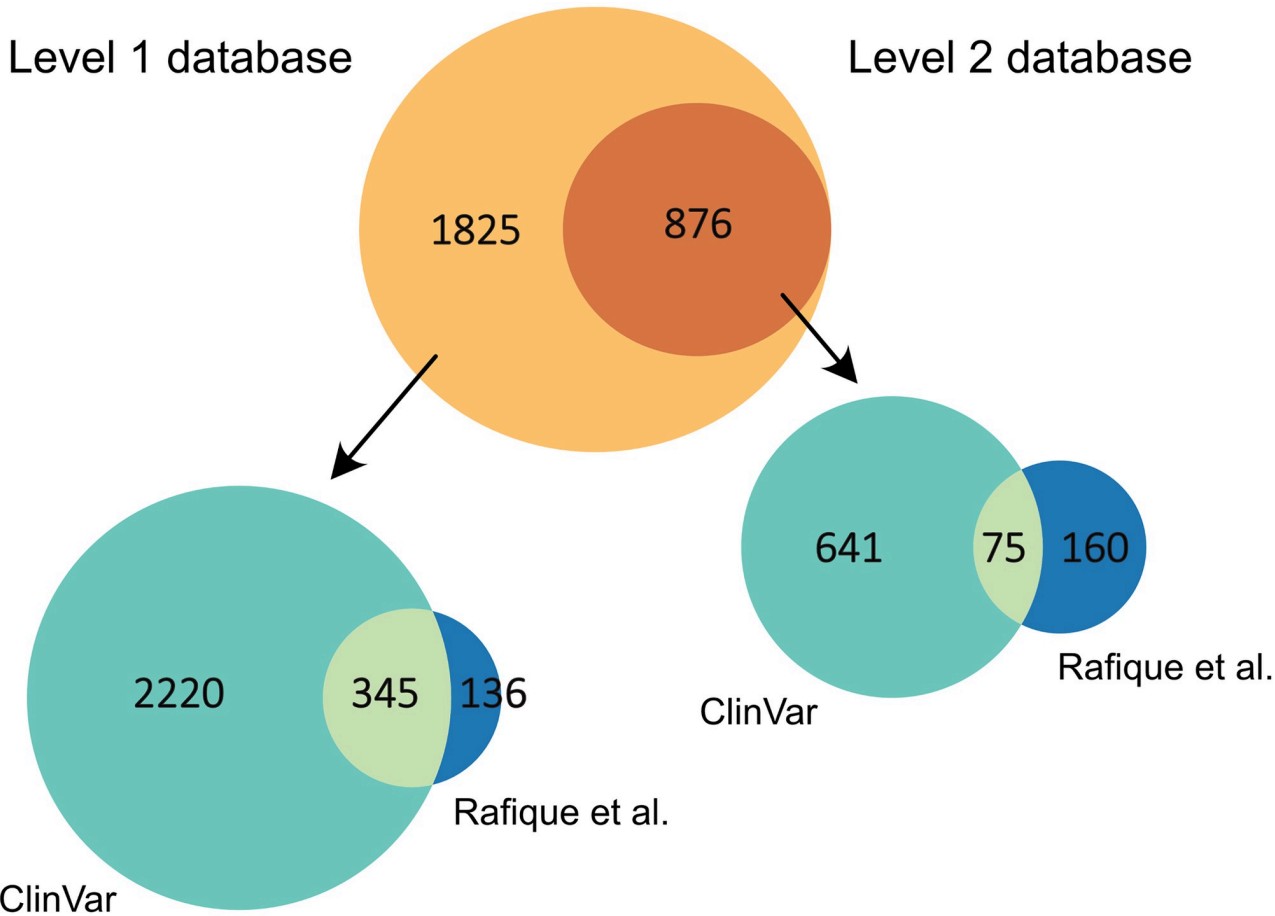

**Fig 4. Venn diagram representing the number of variants in two levels of the database and the number of variants taken from different sources.**
Level 1 database (at the left) consists of i. ClinVar variants retrieved when querying three phenotypes, i.e. "MODY", "monogenic diabetes", and "neonatal diabetes" mapped to Ensembl; ii. All variants from Rafique et al. mapped to Ensembl. Level 2 database (at the right) consists of i. ClinVar "pathogenic" + "likely pathogenic" variants; ii. Variants from Rafique et al. excluded *BLK*, *KLF11*, and *PAX4*.

ClinVar anyway. This is another reason why the overlap of the variants taken from Rafique et al. and ClinVar seems limited in our analysis.

### Distribution of variants among the genes

We observed strong disparities in the number of variants linked to monogenic diabetes among the different genes. *HNF1A*, *GCK*, and *HNF1B* are the genes presenting the most variants in both level 1 and level 2 databases, indicating that most of the monogenic diabetes-associated variants have been reported in these genes, as well as most of the ones confirmed as pathogenic or likely pathogenic (Fig 3). Conversely, genes like *ABCC8* and *KCNJ11* feature many level 1 variants, while only a few of those are confirmed to be pathogenic or likely pathogenic (Fig 3). In fact, of all the genes observed, just about a third bear more than 20 variants, and only around 5% have more than 100 variants. Fig 5 represents the number of gene variants in both levels of the database, where the number of variants in the gene is more than 20. S10 Table of the Supplementary materials and figure "number of variants per gene" in GitHub represent the number of variants in each gene in both levels of the database. Besides the biological reasons that some genes have more association with monogenic diabetes than others, this effect

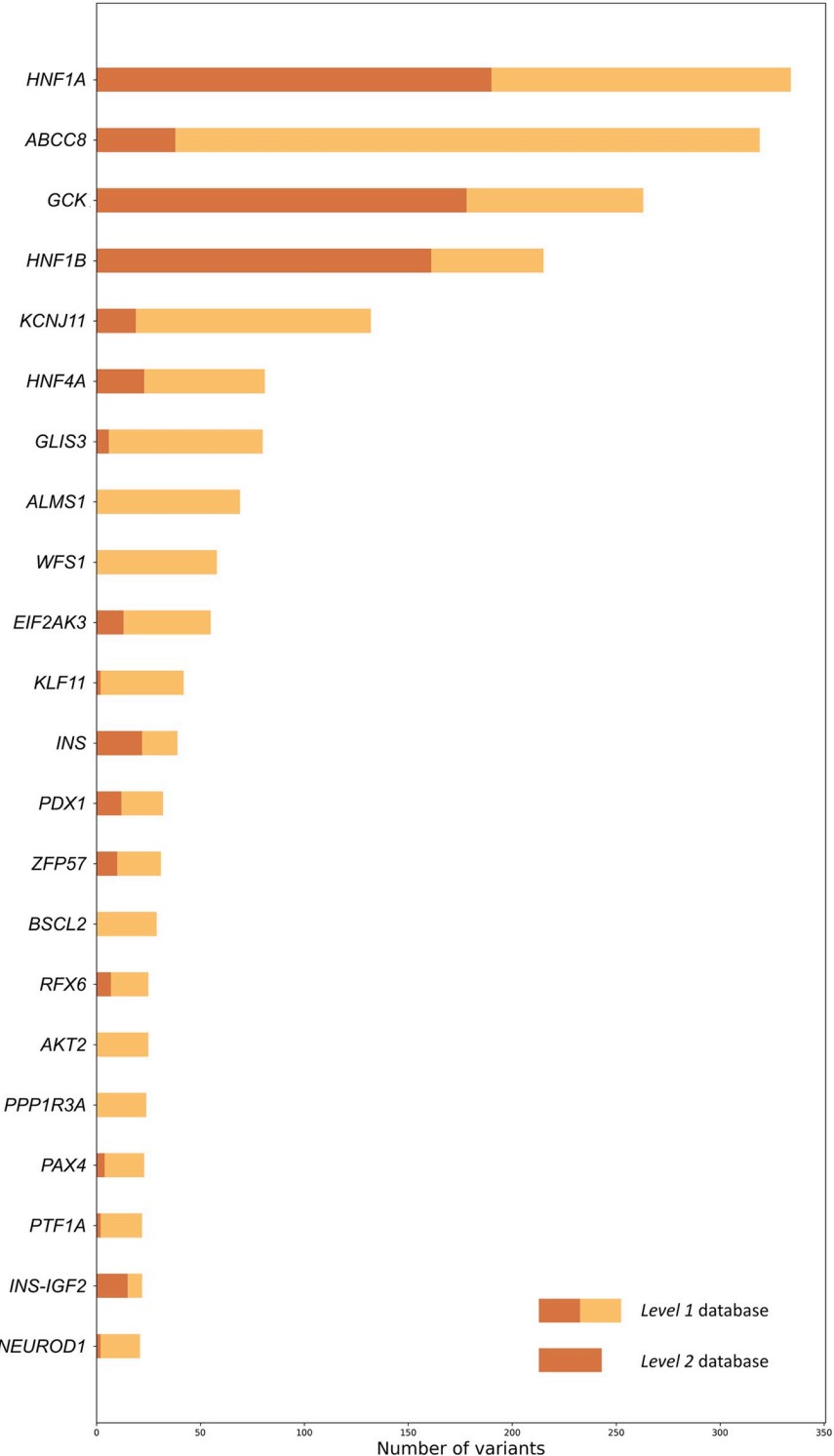

**Fig 5. Number of variants in the genes with the largest number of variants.** Selected are the genes in which the number of variants in the level 1 database is more than 20. The full bars represent the number of variants in the level 1 database, and the brown part of the bars represents the number of variants in the level 2 database.

can be due to study biases or heterogeneous levels of information on these variants. For example, the most common genes known to cause MODY (*HNF1A*, *GCK*, *HNF1B*, and *HNF4A*) [2] might have received more attention than others (e.g. *ABCC8* and *KCNJ11*), yielding a lower confirmation rate for the pathogenicity of the variants. It is also worth noticing that for the time of this manuscript, only variants in *HNF1A*, *GCK*, and *HNF4A* are reviewed by the expert panel on ClinVar (https://clinicalgenome.org/affiliation/40016). At the same time, *ABCC8* is reported to be also associated with other types of diabetes [26] and definitely deserves attention from the perspective of MODY associations as this may bring the researchers closer to the overall understanding of diabetes causal mechanisms and more precise treatment outcome predictions. The variant distribution in the studied genes can be observed in the figures visualizing the variants in the protein sequences (Fig 3 and others in the "figure" directory at (https://github.com/kuznetsovaks/MD_variants).

## Pipeline reproducibility

In order to check whether our pipeline is applicable to other rare diseases and show the reproducibility of the approach, we have run it on Hajdu-Cheney syndrome (HCS). Similar to monogenic diabetes, this is a rare monogenic condition that can be caused by a number of variants in protein-coding regions. All the intermediate files and the results of this analysis can be found in the "HCS" directory at at the GitHub. Finally, we collected all the variants reported to be linked to HCS and highlighted the ones that are classified as pathogenic by ClinVar. All the variant sequences have been translated and added to the human proteome fasta database and the positions of the variants in the protein sequence have been visualized.

## Discussion

In this work, we presented a computational pipeline that allows for systematic monogenic diabetes-linked variant collection and mapping. The sources of information on the genetic variation are not unified which makes mapping of the variants challenging. An automated and reproducible pipeline for variant mapping has been developed and is available for public use. A database of variant protein sequences was created for the gene products of variants linked to monogenic diabetes. All known variants reported to be linked to monogenic diabetes published by the beginning of 2023 have been included in the database. The database contains variants with two levels of clinical significance: variants ever reported as linked to monogenic diabetes and pathogenic variants. Here we were considering the variants pathogenic if they had pathogenic or likely pathogenic clinical significance regardless of the star status according to ClinVar. All the monogenic diabetes-linked variants have been translated into protein products and can be compared to the canonical protein sequences. This will help predict the effect of genetic variation on the resulting protein structure and function.

The workflow is automated and aims to gather multiple variants from different sources and avoid their manual annotation. The implementation in Jupyter notebooks provides a good trade-off between automation and flexibility. For example, researchers can execute the entire pipeline as is, adapt it to specific use cases, execute only modules of interest, or completely change the set of genes to study another disease. The public availability, extensive documentation, and permissive license further enable the reuse of our work.

The collected genetic variants linked to monogenic diabetes have been translated to protein sequences and mapped to all known protein isoforms resulting in the collection of all predicted protein variant sequences. Our database is represented in the form of tables along with FASTA files and can be accessed both manually and automatically allowing implementation in various workflows. The localization of variants on proteins and protein domains

can shed light on their possible consequences and pathogenicity. In turn, overlaying variant pathogenicity on protein sequences can help in understanding protein function. Mutations in protein-coding regions of genes sometimes lead not to a complete stop of expression or protein degradation but rather to structural changes affecting the function of a protein. E.g., in human glucokinase encoded by the GCK gene, sequence variants in a particular region do not affect the catalytic activity of the enzyme. Instead, they increase the rate of degradation and aggregation of the protein, contributing to the molecular mechanism of GCK-MODY disease [27]. The FASTA files can be supplemented with other proteins and used for proteomic search of mass spectrometry data. The variant protein sequences can also be used in protein structure modeling using tools like Alphafold [11] and pathogenicity prediction using such tools as AlphaMissense [28].

This work illustrates how the different genes linked to monogenic diabetes show very different levels of annotation. Besides well-investigated genes, featuring a high number of variants with unambiguous consequences, many understudied genes bear variants lacking evidence of pathogenicity. Finally, after mining different sources, we have created a collection of variants reported to be linked to monogenic diabetes. These variants map to 109 genes. After filtering out all the non-coding variants and the variants in the regions not enriched with pathogenic variants (i.e. untranslated regions etc.) the list of genes consists of 76 genes. Filtering out all the benign and likely benign variants shortens the list of genes down to 36 genes, which, in turn, contains all the 14 previously recognized "MODY genes". All these genes along with the source of the information on their variants and their expert panel review status are presented in Supplementary S10 Table.

Furthermore, some variants simply lack basic genomic annotation, and are reported as amino acid changes, e.g. "Gly292Argfs". An amino acid substitution cannot always be mapped to a single genetic variant. Furthermore, most of the genes encode several protein isoforms and knowing an amino acid change does not give information on which isoform it affects and how it maps to other isoforms. Thus, reporting single amino acid substitutions impairs their inclusion in genomic and bioinformatic studies. In this work, we manually curated these variants and were able to match some of them to genomic coordinates using other variants and aligning protein isoforms using the IsoAligner tool [29]. Moreover, for complex proteomic research in humans, it is important to account for common variation [30]. In future work, we are going to combine our variant sequences with the database of human protein haplotypes [31].

Our work focused on single nucleotide variants (SNVs) or short indels and therefore does not cover larger insertion/deletion mutations causing monogenic diabetes. Large genomic rearrangements, such as full deletion of the HNF1B gene [32] or full deletion of 17q12 locus, have been shown to cause HNF1B-MODY (MODY5) and can be missed by conventional point mutation screening [33, 34]. Insertions or deletions included in our database did not exceed 30 base pairs. These events cannot be directly translated to protein sequences and, as a result, are not reflected in our database.

Furthermore, we would like to emphasize that our work was aimed at creating a pipeline that helps collect variant-level information. While most of the works and reviews provide the clinical context of monogenic diabetes and summarize the collection of genes linked to it, for example, a recent review [34], we collected information on variant positions and allele alternatives in each gene.

In our work, we have analyzed the distribution of pathogenicity among the variants with known consequence types. Based on this analysis we have included variants classified as "splice donor variant", "splice acceptor variant" and filtered out the "splice region variants". In future work, the question of splice region variant consequences should be given more attention as this is an understudied field, and these variants can play a significant role in rare diseases [35].

The research of monogenic diabetes is a dynamically developing field, and new variants are being constantly reported from different cohorts. Now with the pipeline we have developed, we and others can easily update MODY variant collections as new variants are reported. Our pipeline can further be used altogether or in parts to study other diseases. This can enable researchers to automatically and reproducibly collect variants linked to phenotypes of interest and consolidate them to a unified format. In research on rare diseases, the availability of flexible pipelines based on notebooks represents a good compromise between manual expert curation that lacks reproducibility and automated pipelines that cannot be tailored to the application.

## Supporting information

**S1 Table. Exonic variants extracted from Ensembl in Module 1.**
(CSV)

**S2 Table. Result of the 1st step of mapping ClinVar to Ensembl in Module 3.**
(CSV)

**S3 Table. Result of the 2nd step of mapping ClinVar to Ensembl in Module 3.**
(CSV)

**S4 Table. Indels extracted from ClinVar for the level 1 database in Module 3.**
(CSV)

**S5 Table. Indels extracted from ClinVar for the level 2 database in Module 3.**
(CSV)

**S6 Table. Results of the 1st step mapping variants from Rafique et al. to Ensembl in Module 4.**
(CSV)

**S7 Table. Results of the 2nd step mapping variants from Rafique et al. to Ensembl in Module 5.**
(CSV)

**S8 Table. Table ready for VCF file creation for the level 1 database produced in Module 6.**
(CSV)

**S9 Table. Table ready for VCF file creation for the level 2 database produced in Module 6.**
(CSV)

**S10 Table. Summary table of genes and their appearance in different sources.**
(PDF)

## Author Contributions

**Conceptualization:** Ksenia G. Kuznetsova, Marc Vaudel.

**Data curation:** Ksenia G. Kuznetsova, Janne Molnes.

**Funding acquisition:** Marc Vaudel.

**Investigation:** Ksenia G. Kuznetsova.

**Methodology:** Ksenia G. Kuznetsova, Jakub Vašíček, Dafni Skiadopoulou.

**Project administration:** Marc Vaudel.

**Resources:** Marc Vaudel.

**Supervision:** Miriam Udler, Stefan Johansson, Pål Rasmus Njølstad, Alisa Manning, Marc Vaudel.

**Validation:** Jakub Vašíček.

**Visualization:** Ksenia G. Kuznetsova, Alisa Manning.

**Writing – original draft:** Ksenia G. Kuznetsova.

**Writing – review & editing:** Ksenia G. Kuznetsova, Jakub Vašíček, Dafni Skiadopoulou, Janne Molnes, Miriam Udler, Stefan Johansson, Alisa Manning, Marc Vaudel.

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
