## [Decision Letter · Decision Letter 0]

1 Dec 2023

PONE-D-23-29592Systematically mining genomic and proteomic variation linked to rare diseases: the example of monogenic diabetesPLOS ONE

Dear Dr. Kuznetsova,

Thank you for submitting your manuscript to PLOS ONE. After careful consideration, we feel that it has merit but does not fully meet PLOS ONE’s publication criteria as it currently stands. Therefore, we invite you to submit a revised version of the manuscript that addresses the points raised during the review process.

We look forward to receiving your revised manuscript.

Kind regards,

Kazunori Nagasaka

Academic Editor

PLOS ONE

“The work is supported by the Research Council of Norway https://www.forskningsradet.no/

Grant #301178 to M.V.”

“This work was supported by the Research Council of Norway (project #301178 to MV), the

University of Bergen, and the Novo Nordisk Foundation (project NNF20OC0063872 to SJ).

This research was funded, in whole or in part, by the Research Council of Norway 301178. A CC

BY or equivalent license is applied to any Author Accepted Manuscript (AAM) version arising

from this submission, in accordance with the grant’s open access conditions.”

“The work is supported by the Research Council of Norway https://www.forskningsradet.no/

Grant #301178 to M.V.”

Additional Editor Comments:

Dear Authors,

Thank you very much for your submission to Plos One.

I think the manuscript is informative and describes some important points.

Sincerely,

Plos one editorial office

Reviewers' comments:

Reviewer's Responses to Questions

**Comments to the Author**

1. Is the manuscript technically sound, and do the data support the conclusions?

Reviewer #1: Yes

Reviewer #2: Yes

Reviewer #3: Partly

Reviewer #4: Yes

2. Has the statistical analysis been performed appropriately and rigorously? 

Reviewer #1: N/A

Reviewer #2: N/A

Reviewer #3: N/A

Reviewer #4: N/A

3. Have the authors made all data underlying the findings in their manuscript fully available?

Reviewer #1: Yes

Reviewer #2: Yes

Reviewer #3: Yes

Reviewer #4: Yes

4. Is the manuscript presented in an intelligible fashion and written in standard English?

Reviewer #1: Yes

Reviewer #2: Yes

Reviewer #3: Yes

Reviewer #4: Yes

5. Review Comments to the Author

Reviewer #1: The study addresses the issue of pathogenic variation in monogenic diabetes. They developed an automated pipeline to mine the pathogenic variants in over 100 genes considering to be related with monogenic diabetes. By mining the variation data from both ClinVar and literatures and multiple processes, they identified the pathogenic variants in 36 genes, and further sorted out these causing translational changes. All the data from the study are available. The study is certainly a plus for further exploration in the subject.

Comments for improvement:

The definition of short deletions, short insertions, or other short fragment replacements needs to be given and justified. For example, how many bases deleted to be used as the cut off for short deletion?

The use of isoforms in the pipeline can cause mapping trouble as presented by the authors, as the number of isoforms in different genes can be substantial and lack of standardization and quality control. This is the reason to have RefSeq instead of isoforms in genomic annotation. The authors need to justify why use isoform instead of isoforms in their analysis.

“Note that not all these variants could be mapped automatically and those that did not map to Ensembl were formatted manually for input to Module 6. Detailed description needs to be given for formatted “manually”, and indicate the number of the variants under manually check.

ClinVar doesn’t have the classification of “unknown pathogenicity”. It should be the variants of unknown significance or VUS.

“For other types of consequence we focused on those presenting more than 80 % pathogenic and likely pathogenic variants”. Justification for 80% needs to be given, and the number at this cut-off needs to be given.

“We could map 2,701 of these variants to variants linked to monogenic diabetes according to

ClinVar or the literature”: should be “We mapped 2701 of….”

The version numbers for Ensembl, ClinVar and dbSNP need to be indicated. Different versions can be a reason for the inconsistence of the mapping results, such as “why the overlap of the variants taken from Rafique et al. and ClinVar seems limited in our analysis”.

The value of translation products has been repeatedly indicated to justify the development of translational products, as it “help predict the effect of genetic variation on the resulting protein structure and function”. However, there is no single example to justify their claim. I would suggest to present an example of coding changed protein with structural alteration as a proof of principle, considering they have generated rich coding-change pathogenic variants in multiple genes, like HNF1A.

The resolution is poor for all figures.

Reviewer #2: The authors present an interesting paper on the computational assessment of coding variants on protein sequences, categorizing these variants into different tiers depending on predicted pathogenicity.

Some minor comments to the authors

- The authors use the article by Rafique et al (2021). However, they should also mention the paper published by Bonnefond et al (2023) Nature Reviews particularly in their Discussion section (page 14) and how this compares to what they included in their study.

- Pg 4: Change port to import

- Page 3 “It is estimated that around 77 % of monogenic diabetes cases remain undiagnosed”. Specify why.

- Pg 3: Perhaps specify the total number of genes (to date) related to monogenic diabetes not just the 14 MODY genes

- I think it should be specified before (on page 5) which type of exonic variants were retained (listed on page 6)

- It would be good to specify the list of in silico tools used by Ensembl (VEP) to predict pathogenicity

- Did the authors consider cross checking some of the level 1/2 variants with ACMG/AMP predictions?

- Did the authors apply an alternative allele frequency cut-off? Perhaps some of the e.g. nonsense variants might be common.

- Page 8: Variants in BLK, KLF11 and PAX4 should be removed for dominant monogenic diabetes and it is still nonetheless considered controversial.

- Supp Table 5: would change to Yes/No in Rafique column

- Perhaps Links should be included as footnotes (rather than references in text)

Reviewer #3: This manuscript provides a modular Jupiter notebook approach for collecting information on monogenic diabetes variants across 108 genes. The authors compile their database from ClinVar and literature to generate a uniform list with annotations.

The work presented is thorough, though I was unable to personally validate the coding implementation. Additionally, the authors did not demonstrate this approach generalizes well to other diseases, making it difficult to assess wider utility.

Some of the approaches rely on circular logic - for example, the variants in Module 1 already include ClinVar data and use the Ensemble Variant Effect Predictor, which provides comprehensive annotations.

While this tool aims to fill a need, numerous established resources (DECIPHER, VEP etc.) and their continuous expansions already provide rich variant annotation and data integration. As more addons targeting monogenic diseases are actively developed and included in these tools actively the niche for this specific tool may diminish over time due to manual update needed for this work.

Overall, this manuscript describes a thoughtful effort to aggregate information on monogenic diabetes variants. However, given the breadth of existing tools, I am unable to clearly recognize the unmet need this resource fills in the community.

Reviewer #4: Authors developed automated and reproducible bioinformatics pipeline for mapping variants in genes related to MODY, and further collecting and harmonizing data about them. Their work addresses current challenges that geneticist face in the process of interpreting effect of rare/new genetic variants.

The pipeline has been described in detail and is made available for public use.

Additional value is that parts of this pipeline could be used to develop similar pipelines for other rare genetic disorders / groups of rare disorders. This is a needed tool in the rare disease ecosystem.

Therefore, I highly recommend publishing this manuscript in the current form.

Minor comment: I suggest including words “bioinformatics pipeline” in the title to facilitate bioinformaticians to find this valuable manuscript.

6. PLOS authors have the option to publish the peer review history of their article (what does this mean?). If published, this will include your full peer review and any attached files.

Reviewer #1: No

Reviewer #2: No

Reviewer #3: No

Reviewer #4: **Yes: **Maja Stojiljkovic

---

## [Author Response · Author response to Decision Letter 0]

30 Jan 2024

We thank the editor and all the reviewers for their attention to our work and valuable comments. We have implemented all suggestions and as a result the manuscript is substantially improved. 

We are providing the revised version in the form of a LaTex document as it was requested by the journal and also attaching the MS Word file with the changes highlighted. 

We have not made any changes in the figures, but have added one supplementary table (S10_table), renamed the supplementary files, and updated all the in-text citations as required. 

We have modified the reference list and added 5 more references (28, 30, 31, 32, 34). The updated reference list is included at the end of the LaTex file as required by the template. 

We have added a license file to the GitHub repository and a file containing all the dependencies for creating the conda environment for running our pipeline with explanations in a readme file.

Reviewer #1:

We thank reviewer 1 for the thorough examination of our work and for their appreciation of its contribution to monogenic disease research. We have implemented all suggestions, as detailed in the answers below.

The definition of short deletions, short insertions, or other short fragment replacements needs to be given and justified. For example, how many bases deleted to be used as the cut off for short deletion?

Filtering the variants we did not set any limits deliberately, but out of the variants from ClinVar linked to monogenic diabetes, the indels were no longer than 30 bp. We have added this information to page 17.

The use of isoforms in the pipeline can cause mapping trouble as presented by the authors, as the number of isoforms in different genes can be substantial and lack of standardization and quality control. This is the reason to have RefSeq instead of isoforms in genomic annotation. The authors need to justify why use isoform instead of isoforms in their analysis.

In our study, we translated all variant genes into the possibly encoded proteins to enable protein-level analyses (e.g. structural or mass spectrometry-based analysis). In order to comprehensively cover all possibly encoded proteins, we chose to include all possible isoforms as provided by Ensembl. The reviewer is correct that analysts usually reduce the number of isoforms and in most cases use only canonical sequences, as provided by RefSeq or Ensembl. In the study of rare diseases, and monogenic diabetes in particular, it is important to be able to select between isoforms. For example, different isoforms of GCK are specific for different tissues https://www.degruyter.com/document/doi/10.1515/hsz-2018-0109/html?lang=de. It is therefore valuable for analysts to have all isoforms available and choose the ones relevant to their analysis, instead of being limited to canonical isoforms. This has been clarified in the text.

“Note that not all these variants could be mapped automatically and those that did not map to Ensembl were formatted manually for input to Module 6. Detailed description needs to be given for formatted “manually”, and indicate the number of the variants under manually check.

The description is added to page 8.

ClinVar doesn’t have the classification of “unknown pathogenicity”. It should be the variants of unknown significance or VUS.

We apologize for the incorrect formulation, it has been corrected with the denomination from ClinVar as suggested.

“For other types of consequence we focused on those presenting more than 80 % pathogenic and likely pathogenic variants”. Justification for 80% needs to be given, and the number at this cut-off needs to be given.

“We could map 2,701 of these variants to variants linked to monogenic diabetes according to

ClinVar or the literature”: should be “We mapped 2701 of….”

This has been corrected on page 10.

The version numbers for Ensembl, ClinVar and dbSNP need to be indicated. Different versions can be a reason for the inconsistence of the mapping results, such as “why the overlap of the variants taken from Rafique et al. and ClinVar seems limited in our analysis”.

This has been added.

The value of translation products has been repeatedly indicated to justify the development of translational products, as it “help predict the effect of genetic variation on the resulting protein structure and function”. However, there is no single example to justify their claim. I would suggest to present an example of coding changed protein with structural alteration as a proof of principle, considering they have generated rich coding-change pathogenic variants in multiple genes, like HNF1A.

We have added an example of structural changes in GCK due to sequence coding region mutation from doi.org/10.1016/j.bbadis.2012.07.005 and variant pathogenicity prediction using AlphaMissense that is based on the structural predictor AlphaFold doi/epdf/10.1126/science.adg7492 on page 16. We thank the reviewer for this suggestion.

The resolution is poor for all figures.

We have provided high-resolution figures as separate files along with the manuscript as well as on GitHub. Unfortunately, it seems that the compression of the submission system altered the resolution. We will ensure that resolution is sufficient in the resubmission, alternatively, all the files are available for download from https://github.com/kuznetsovaks/MD_variants

Reviewer #2:

We thank reviewer 2 for the careful analysis of our work and for pointing out our approach to dividing variants into tiers. We have implemented all suggestions, as detailed in our answers below.

- The authors use the article by Rafique et al (2021). However, they should also mention the paper published by Bonnefond et al (2023) Nature Reviews particularly in their Discussion section (page 14) and how this compares to what they included in their study.

We thank the reviewer for pointing out this important paper. We have now included this this review in the paper (See page 17 and Supplementary Table 10). However, unlike the work by Rafique et al., this manuscript provides an overview of MG at the gene level and lists the linked genes but not the variants. We have checked the overlap of the gene list in the review with the gene list in our work. As expected, the overlap is not perfect, but discussing the differences between the gene lists provided by different reviews is beyond the scope of our work.

- Pg 4: Change port to import

This has been corrected.

- Page 3 “It is estimated that around 77 % of monogenic diabetes cases remain undiagnosed”. Specify why.

Thank you for this suggestion, we have extended the text accordingly.

- Pg 3: Perhaps specify the total number of genes (to date) related to monogenic diabetes not just the 14 MODY genes

One of the points and conclusions of our study is that the list of genes related to monogenic diabetes is a matter of continuous discussion and revision. A gene can be included or excluded from this list based on the pathogenicity, frequency etc. of its genetic variants, which in turn is constantly being revised. We have extended the result section and discussion accordingly, but for the sake of clarity the introduction does not expand on these considerations. 

- I think it should be specified before (on page 5) which type of exonic variants were retained (listed on page 6)

This has been added, thank you for the suggestion.

- It would be good to specify the list of in silico tools used by Ensembl (VEP) to predict pathogenicity

This has been specified.

- Did the authors consider cross checking some of the level 1/2 variants with ACMG/AMP predictions?

The main aim of our work was to collect, standardize, and translate the variants linked to monogenic diabetes using open-source and freely available tools and databases. There are many different tools for ACMG/AMP classification such as older ones (doi: 10.1186/s13059-017-1353-5) and newer machine learning-based ones (doi: 10.1038/s41598-022-06547-3) as well as commercial tools. To our understanding, these classifiers are designed to be used on datasets of variants called from actual sequencing data rather than genomic knowledgebases. We find this analysis an important and interesting part of the patient classification framework but it is outside the scope of our study. 

- Did the authors apply an alternative allele frequency cut-off? Perhaps some of the e.g. nonsense variants might be common.

The main goal of our work was to collect and standardize the information on the possible protein alterations connected to monogenic diabetes. For this reason, we based our variant selection primarily on the variant location (exonic variants), secondarily on protein consequences (missense SNP, frameshift etc.), and thirdly on reported pathogenicity. For the latter, we separated our database into two levels: one including all the variants and the second one including just the pathogenic ones. This approach is implemented to address the limitations of the ‘street light effect’ appearing in mass spectrometry-based proteomics. Since the proteomics spectra are analyzed against a protein sequence database, the sequences not included in the database have no chance of being found in the samples. For this reason, we tried to keep the level 2 database as thorough as possible and create a strict version of the database limited to only pathogenic variants. Here we did not use any allele frequency cut-off. Nevertheless, we controlled for the alternative allele frequencies of the variants for which they were available using the Ensemple REST API and see that the level 2 database contains approximately 8.5% of common and 11% of low-frequency variants while the level 1 database contains 0.3% of low frequency variants and no common variants. We did not include this analysis in the manuscript as we believe that it will introduce complexity and distract attention from the main purpose of our study. 

- Page 8: Variants in BLK, KLF11 and PAX4 should be removed for dominant monogenic diabetes and it is still nonetheless considered controversial.

We thank the reviewer for this suggestion, these genes have been discussed at length in our lab during the past years. We agree with the reviewer and have removed them from the level 2 database but kept them in level 1, as level 1 is the most complete collection of variants that have been reported. We mention the controversy on these three genes and cite the paper discussing it on page 12.

- Supp Table 5: would change to Yes/No in Rafique column

This has been corrected

- Perhaps Links should be included as footnotes (rather than references in text)

This has been corrected in the LaTex version following the template of the journal.

Reviewer #3: 

We thank reviewer 3 for their critical review of our work. We have implemented all suggestions as detailed in the comments below.

The work presented is thorough, though I was unable to personally validate the coding implementation. Additionally, the authors did not demonstrate this approach generalizes well to other diseases, making it difficult to assess wider utility.

We have reproduced the pipeline on an example of a Hajdu-Cheney syndrome. All the files can be found in GitHub in the "HCS" directory. The text has been extended accordingly (see page 14).

Some of the approaches rely on circular logic - for example, the variants in Module 1 already include ClinVar data and use the Ensemble Variant Effect Predictor, which provides comprehensive annotations.

While this tool aims to fill a need, numerous established resources (DECIPHER, VEP etc.) and their continuous expansions already provide rich variant annotation and data integration. As more addons targeting monogenic diseases are actively developed and included in these tools actively the niche for this specific tool may diminish over time due to manual update needed for this work.

Overall, this manuscript describes a thoughtful effort to aggregate information on monogenic diabetes variants. However, given the breadth of existing tools, I am unable to clearly recognize the unmet need this resource fills in the community.

We are sorry to read that we failed to convey how our pipeline answers a need from the biomedical community. The main aim of our work is to collect genetic variants linked to monogenic information and translate them into protein sequences. While we acknowledge the value of resources like VEP, DECYPHER, and ClinVar, the nature of rare diseases makes it challenging for generalist solutions to stay updated with field-specific advances. When creating proteomic databases enhanced with genetic data to analyze patients' proteomes, the community currently often relies on manual sequence analysis. While we share the optimism of the reviewer that specialist addons will be included, eventually streamlining the work on rare diseases, we believe that to date, semi-automated approaches like ours provide a good compromise between exhaustivity and reproducibility. 

Reviewer #4: 

We thank Prof. Stojiljković for their appreciation of our contribution and a valuable suggestion to change the title. We have changed it accordingly: “Bioinformatics pipeline for the systematic mining genomic and proteomic variation linked to rare diseases: the example of monogenic diabetes”

---

## [Decision Letter · Decision Letter 1]

27 Feb 2024

Bioinformatics pipeline for the systematic mining genomic and proteomic variation linked to rare diseases: the example of monogenic diabetes

PONE-D-23-29592R1

Dear Dr. Kuznetsova,

We’re pleased to inform you that your manuscript has been judged scientifically suitable for publication and will be formally accepted for publication once it meets all outstanding technical requirements.

Kind regards,

Kazunori Nagasaka

Academic Editor

PLOS ONE

Additional Editor Comments (optional):

Dear Authors,

Thank you so much for submitting your manuscript to JOGR.

Now your manuscript is acceptable for publication in Plos One.

We look forward to your future submission.

Sincerely,

Kazunori Nagasaka

Reviewers' comments:

Reviewer's Responses to Questions

**Comments to the Author**

1. If the authors have adequately addressed your comments raised in a previous round of review and you feel that this manuscript is now acceptable for publication, you may indicate that here to bypass the “Comments to the Author” section, enter your conflict of interest statement in the “Confidential to Editor” section, and submit your "Accept" recommendation.

Reviewer #1: All comments have been addressed

Reviewer #2: All comments have been addressed

Reviewer #4: All comments have been addressed

2. Is the manuscript technically sound, and do the data support the conclusions?

Reviewer #1: Yes

Reviewer #2: Yes

Reviewer #4: Yes

3. Has the statistical analysis been performed appropriately and rigorously? 

Reviewer #1: N/A

Reviewer #2: Yes

Reviewer #4: N/A

4. Have the authors made all data underlying the findings in their manuscript fully available?

Reviewer #1: Yes

Reviewer #2: Yes

Reviewer #4: Yes

5. Is the manuscript presented in an intelligible fashion and written in standard English?

Reviewer #1: Yes

Reviewer #2: Yes

Reviewer #4: Yes

6. Review Comments to the Author

Reviewer #1: The revision has addressed my questions with satisfaction. The quality of revision is substantially improved over the original version.

Reviewer #2: (No Response)

Reviewer #4: The authors made changes in line with the first round of the review. I do not have any further comments.

7. PLOS authors have the option to publish the peer review history of their article (what does this mean?). If published, this will include your full peer review and any attached files.

Reviewer #1: **Yes: **San Ming Wang

Reviewer #2: No

Reviewer #4: **Yes: **Maja Stojiljkovic

---

## [Editor Report · Acceptance letter]

25 Mar 2024

PONE-D-23-29592R1 

PLOS ONE

Dear Dr. Kuznetsova, 

I'm pleased to inform you that your manuscript has been deemed suitable for publication in PLOS ONE. Congratulations! Your manuscript is now being handed over to our production team.

Kind regards, 

on behalf of

Professor Kazunori Nagasaka 

Academic Editor

PLOS ONE